# Novel High-Entropy FeCoNiMoZn-Layered Hydroxide as an Efficient Electrocatalyst for the Oxygen Evolution Reaction

**DOI:** 10.3390/nano14100889

**Published:** 2024-05-20

**Authors:** Zhihao Cheng, Xin Han, Liying Han, Jinfeng Zhang, Jie Liu, Zhong Wu, Cheng Zhong

**Affiliations:** 1Key Laboratory of Advanced Ceramics and Machining Technology (Ministry of Education), and Tianjin Key Laboratory of Composite and Functional Materials, School of Materials Science and Engineering, Tianjin University, Tianjin 300072, China; 18002171015@163.com (Z.C.); 18822144032@163.com (X.H.); 18822029109@163.com (L.H.); jinfeng@tju.edu.cn (J.Z.); zhong.wu@tju.edu.cn (Z.W.); cheng.zhong@tju.edu.cn (C.Z.); 2Joint School of National University of Singapore and Tianjin University, International Campus of Tianjin University, Binhai New City, Fuzhou 350207, China

**Keywords:** high entropy, water splitting, oxygen evolution reaction, electrocatalytic

## Abstract

The exploration of catalysts for the oxygen evolution reaction (OER) with high activity and acceptable price is essential for water splitting to hydrogen generation. High-entropy materials (HEMs) have aroused increasing interest in the field of electrocatalysis due to their unusual physicochemical properties. In this work, we reported a novel FeCoNiMoZn-OH high entropy hydroxide (HEH)/nickel foam (NF) synthesized by a facile pulsed electrochemical deposition method at room temperature. The FeCoNiMoZn-OH HEH displays a 3D porous nanosheet morphology and polycrystalline structure, which exhibits extraordinary OER activity in alkaline media, including much lower overpotential (248 mV at 10 mA cm^−2^) and Tafel slope (30 mV dec^−1^). Furthermore, FeCoNiMoZn-OH HEH demonstrates excellent OER catalytic stability. The enhanced catalytic performance of the FeCoNiMoZn-OH HEH primarily contributed to the porous morphology and the positive synergistic effect between Mo and Zn. This work provides a novel insight into the design of HEMs in catalytic application.

## 1. Introduction

Hydrogen with high combustion efficiency and green renewability has been widely recognized as a promising and achievable new energy source to replace fossil fuel [1,2]. Electrochemical water splitting is a highly efficient hydrogen production technology without polluting byproducts [3,4]. Nevertheless, as one of the key reactions of water electrolysis, the oxygen evolution reaction (OER) suffers from high overpotential and sluggish reaction kinetics as a result of the complicated four-electron transfer process [5,6,7]. Currently, noble metal-based electrocatalysts, including RuO_2_ and IrO_2_, are considered excellent electrocatalysts for OER [3,5,8]. However, their scarcity, high price, poor durability, and low selectivity restricted their large-scale commercial application [9,10,11]. Consequently, it is of great importance to develop non-noble metal electrocatalysts with acceptable cost, high efficiency, and satisfactory durability for OER.

In this scenario, catalysts composed of transition metal have been widely studied as promising OER catalysts because of their abundant resources and low price. To date, various electrocatalysts consisting of the transition metal have been extensively developed, including NiFe [12], NiFe hydroxide [13], FeCoW hydroxide [14], and NiFeMo oxide [15]. However, the binary or ternary catalysts lack broad compositional tunability [16], resulting in a limited number and types of active sites [17].

Recently, high-entropy materials (HEMs), comprised at least five principal elements, have aroused extensive interest in the field of electrocatalysis as a result of their unusual physicochemical properties. The large lattice distortion in HEMs not only induces plenty of lattice defects on the surface, which allows the introduction of plentiful active sites but also limits the atomic diffusion to enhance the structural stability. Furthermore, the synergistic interaction among multiple principle components improves the tunability of the electronic structure of HEMs [6,18,19,20]. Therefore, HEM electrocatalysts have been extensively studied in the past few years. Recently, Huang et al. [21] synthesized FeCoNiMnCu high entropy alloy (HEA) using a solvothermal method; the synthesized HEA catalyst exhibited outstanding catalytic activity and stability for catalyzing OER. Strotkoetter et al. [22] fabricated the FeCoNiCrMn high entropy oxide via the codeposition method and this catalyst showed an outstanding OER electrocatalytic performance in an alkaline medium. So far, various methods have been developed to synthesize HEM catalysts, including plasma sintering [23], arc-melting [24], carbothermal shock [25], and laser ablation [26]. Nevertheless, most of these methods usually involve harsh high-temperature conditions [27,28]. Consequently, it is highly desirable to explore a facile technique under mild atmospheres for the synthesis of HEM catalysts with high activity and excellent durability [29]. Electrodeposition, as a conventional method, has been extensively used to fabricate electrocatalysts owing to its low cost and simple process. For example, He et al. [30] synthesized FeCoNiMn (oxy) hydroxide by electrodeposition for OER catalysis. Bian et al. [31] prepared FeMnCuCo HEA as an efficient OER catalyst via one-step electrodeposition.

It is noteworthy that the performance of high entropy catalysts is strongly dependent on component selection. The adjustable elemental components and ratios of high entropy catalysts offer a possibility to develop catalysts with higher catalytic activity [19]. Especially, the catalysts based on 3d-transition metals including Fe, Ni, Co, etc., demonstrated excellent electrocatalytic performance in alkaline media [19,32]. Furthermore, Zn could be combined with active metal Ni to modulate the electronic property of the catalyst [6,33]. For instance, Huang et al. [34] synthesized the Fe_0.5_CoNiCuZn_x_ catalyst by electro-deoxidizing molten salts at high temperature, this electrode shows a low overpotential (340 mV at 10 mA cm^−2^) and superior durability for 24 h. The introduction of Zn alters the HEA bond with the element and thereby increases the adsorption energies between the surface of the HEA catalyst and OH^−^/H^+^. Moreover, the introduction of Zn also increases the interplanar spacing of HEA and thus enhances the OER activity [34]. Additionally, as a 4d high-valence metal, Mo has a larger spatial extent, being endowed with strong hybridization with adjacent ligand orbitals. The large electronic bandwidth of Mo allows for a wider tunable range of electronic structures [35]. Furthermore, Mo metals have additional orbital degrees of freedom, which facilitates the tuning of the electronic band structure and the adsorption/desorption energy of the intermediates [32]. For instance, Qiu et al. [36] developed the AlNiCoRuMo HEA by a melting and de-alloying process. The catalyst exhibited an outstanding OER activity. The addition of Mo makes the e_g_-orbital occupation of both Co and Ru close to unity, which results in a high OER of the catalyst. Li et al. [37] designed a FeCoNiMnMo HEA by molten salt electrolysis of oxides and this catalyst showed lower overpotentials (279 mV at 10 mA cm^−2^) and smaller tafel slopes (56.1 mV dec^−1^). However, the combination of Fe, Co, Ni, Mo, and Zn in OER catalysis has not already been reported and its interactions are not clear. Considering that both Mo and Zn have promotive effects on OER electrocatalysis, it is therefore possible that the combination of Mo and Zn may have positive synergistic effects on FeCoNiMoZn catalysts. 

Herein, we synthesized the FeCoNiMoZn-OH-layered high entropy hydroxide (HEH) on the surface of nickel foam (NF) via the pulse electrodeposition method under a mild atmosphere. The obtained FeCoNiMoZn-OH HEH exhibits a three-dimensional (3D) porous morphology, which consists of a large number of nanosheets. This unique structure ensures an ample specific surface area and facilitates the ions transfer. The obtained FeCoNiMoZn-OH catalyst afford outstanding OER catalytic performance under alkaline condition, including a lower overpotential of 248 mV at 10 mA cm^−2^, a smaller tafel slope of 30 mV dec^−1^, and a long-term durability of 55 h compared with the commercial RuO_2_.

## 2. Experimental Section

### 2.1. Reagents and Materials

Ferrous (II) sulfate heptahydrate (FeSO_4_·7H_2_O, 99%), sodium dodecylsulfate (SDS), hydrochloric acid (HCl, 37%), and anhydrous athanol (C_2_H_5_OH, 99.7%) were purchased from Feng Chuan Chemical Reagent Co., Ltd. (Tianjin, China). Cobalt sulfate heptahydrate (CoSO_4_·7H_2_O, 99.5%) was bought from Macklin Biochemical Co., Ltd. (Shanghai, China). Isopropyl alcohol (C_3_H_8_O, >99%) and nickel sulfate hexahydrate (NiSO_4_·6H_2_O, 99%) was obtained from Aladdin Chemical Technology Co., Ltd. (Shanghai, China). Sodium molybdate dihydrate (Na_2_MoO_4_·2H_2_O, ≥99.5%) was obtained from Dibo Chemicals Technology Co., Ltd. (Shanghai, China) Zinc(II) sulfate heptahydrate (ZnSO_4_·7H_2_O), trisodium citrate dihydrate (Na_3_C_6_H_5_O_7_·2H_2_O, ≥99.5%), and potassium hydroxide (KOH, 85%) were supplied by Kermel Chemical Reagent Co., Ltd. (Tianjin, China). Ruthenium (IV) oxide (RuO_2_, 97%) was bought from Haohong Scientific Co., Ltd. (Shanghai, China). Nickel foam (NF) was obtained from Leviathan Technology Co., Ltd. (Tianjin, China).

### 2.2. Synthesis of HEH Catalyst

FeCoNiMoZn-OH HEH was synthesized on the surface of NF by a pulse electrodeposition method. Firstly, a NF (1mm thick, 110 ppi, exposed area of 1 cm^2^) was ultrasonically cleaned in hydrochloric acid, ethanol, and deionized water (DI). Secondly, a conventional three-electrode system was used to prepare the FeCoNiMoZn-OH HEH catalyst. The working electrode was the cleaned NF. The reference electrode and counter electrode were the Ag/AgCl electrode and graphite rod, separately. The distance between the working electrode and the counter electrode was 2.50 cm. In total, 0.04 M FeSO_4_, 0.04 M CoSO_4_, 0.05 M NiSO_4_, 0.025 M Na_2_MoO_4_, 0.03 M ZnSO_4_, 0.1 M Na_3_C_6_H_5_O_7_, and 0.15 g L^−1^ SDS were dissolved sequentially into an aqueous solution to obtain the electrolyte. The FeCoNiMoZn HEH was electrodeposited by square pulse potential mode at 25 °C using an electrochemical workstation (CHI 760E, Chenhua Instruments, Inc., Shanghai, China). The upper potential limit and pulse duration were 0 V and 0.1 s; the lower potential limit and pulse duration were set as −2 V and 0.1 s, respectively. The entire electrode position process lasted for 3000 cycles. Thirdly, the obtained electrode was cleaned with DI water repeatedly. FeCoNi-OH, FeCoNiMo-OH, and FeCoNiZn-OH catalysts were prepared following a similar process to FeCoNiMoZn-OH HEH. The concentrations of the corresponding metal salts were kept consistent throughout the synthesis.

### 2.3. Materials Characterization

The morphology of FeCoNiMoZn-OH HEH was characterized by a field-emission scanning electron microscope (FE-SEM, JSM-7800F, JEOL, Tokyo, Japan) and a high-resolution transmission electron microscope (HRTEM, JEM-F200, JEOL, Tokyo, Japan) equipped with energy-dispersive X-ray spectroscopy (EDX). The phase characteristics of the sample were investigated via the X-ray powder diffractometer (XRD, Bruker D8 Advanced, Ettlingen, Germany). The chemical states of the obtained catalyst elements were detected via X-ray photoelectron spectroscopy (XPS, AXIS SUPRA, Kratos, Manchester, UK) using an Al Kα X-ray as the exciting source. The coordination environment as well as functional group within the material were characterized by Raman spectroscopy (Thermo Fischer DXR, Waltham, MA, USA) and Fourier transform infrared spectroscopy (FTIR, Thermo Fisher is 10, Waltham, MA, USA).

### 2.4. Electrochemical Tests

In this work, OER catalytic performance tests were conducted with a standard three-electrode system (the electrochemical workstation, CHI 760E). The working electrode was NF-loaded with catalysts and the counter electrode and reference electrode were Pt foil and Ag/AgCl, separately. The electrolyte was 1.0 M KOH aqueous solution. All potentials (vs. Ag/AgCl) were converted into potentials against the reversible hydrogen electrode (RHE) (E_RHE_ = E_Ag/AgCl_ + 0.059 pH + 0.197 V). Before all electrochemical testing, cyclic voltammetry (CV) was conducted for 20 cycles at 40 mV s^−1^ to energize the catalyst activity. Linear sweep voltammetry (LSV) was employed to measure the catalytic efficiency for the OER of the samples. The LSV polarization curves ranged from 1.2 to 1.7 V (vs. RHE) and were collected at 5 mV s^−1^. Tafel slopes of these samples were determined from the LSV curves (the linear region). Tafel slopes were estimated according to equation *η* = b log *j* + a, where *η* is the overpotential of the electrode, *j* is the current density, b is the tafel slope, and a is the intercept of the fitted curve. All the polarization potentials obtained in the LSV test were implemented with iR compensation (90%). Electrochemical impedance spectroscopy (EIS) tests were performed in a KOH solution at the overpotential corresponding to the current density of 10 mA cm^−2^. The lower- and upper-frequency limits during the test were 0.01 Hz and 100 kHz, separately. The obtained catalysts were subjected to CV tests in 1.0 M KOH solution at different scan rates and CV curves ranging from 0.72 to 0.82 V (vs. RHE) were collected to evaluate the electrochemical active surface area (ECSA). The ECSA of the samples were evaluated by the equation ECSA = C_dl_/C_s_, where C_dl_ (double-layer capacitance) was obtained by plotting Δ*j* = (*j_a_* − *j_c_*) against the scan rates, which is half of the value of the linear slope. C_s_ (specific capacitance) was set as 40 μF cm^−2^ [38]. The long-term durability was evaluated by chronopotentiometry tests. For comparison, the benchmark RuO_2_ electrodes (2.5 mg cm^−2^) were fabricated. Overall, 10 mg of commercial RuO_2_ powder, 35 μL Nafion (5 wt. % in ethanol), and 965 μL of isopropanol were mixed and then ultrasounded for 30 min, followed by the formation of a homogeneous ink suspension. Then, 250 μL of this suspension was evenly drop-coated on the surface of NF and the sample was finally dried. In all testing, each set of tests is performed more than three times to ensure that the obtained data are within the permissible range of error.

## 3. Results and Discussion

Compared with the smooth morphology of bare nickel foam (Appendix A), the surface of the deposited NF is entirely covered by the FeCoNiMoZn-OH HEH catalyst layer (Figure 1a,b). The HEH layer exhibits the 3D porous morphology, which consists of a large number of nanosheets (Figure 1b). This unique 3D porous structure not only favors the provision of ample specific surface area but also promotes ions/mass transportation. Figure 1c displays the XRD spectrum of the FeCoNiMoZn-OH HEH. The main diffraction peaks appear at 19.3°, 33.1°, 38.4°, 51.8°, 60.2°, 70.8°, and 81.2°, indexing to (001), (100), (002), (012), (111), (201), and (202) of FeCoNiMoZn-OH HEH (JCPDS 74-1057). This indicates that the FeCoNiMoZn-OH HEH has a polycrystalline structure. To further investigate the microstructure of FeCoNiMoZn-OH HEH, Figure 1d shows a TEM image of FeCoNiMoZn-OH HEH. The HEH possesses the ultrathin nanosheet structure, which is in accordance with the morphology shown in SEM (Figure 1b). The HRTEM of FeCoNiMoZn-OH HEH (Figure 1e) displays the interplanar spacing of 0.15, 0.20, and 0.23 nm (Figure 1e), indicating (111), (202), and (002) planes of FeCoNiMoZn-OH HEH, respectively. Also, FeCoNiMoZn-OH HEH displays a large number of lattice defects, demonstrating the lattice distortion effect of high entropy materials. The red and green symbols indicate the dislocation and stacking faults, separately. The selected area electron diffraction (SAED) pattern exhibits multiple sets of bright diffraction concentric rings (Figure 1f), further confirming the polycrystalline structure of FeCoNiMoZn-OH HEH. Furthermore, the EDS line and mapping spectra of FeCoNiMoZn-OH HEH show the existence of Fe, Co, Ni, Mo, Zn, and O elements (Appendix A; Figure 1g) and these elements are uniformly distributed on the whole sample (Figure 1g). The atomic weight percentage of the corresponding elements is also listed in Appendix A.

An XPS test was conducted to characterize the chemical states of FeCoNiMoZn-OH HEH. The XPS survey spectra of FeCoNiMoZn-OH HEH present the peaks of Fe 2p, Co 2p, Ni 2p, Mo 3d, Zn 2p, and O 1s (Appendix A). The high-resolution spectrum of Fe 2p (Figure 2a) shows peaks at 706.2 eV, 708.8 eV, and 711.9 eV, corresponding to Fe^0^ 2p_3/2_, Fe^2+^ 2p_3/2_, and Fe^3+^ 2p_3/2_. The peaks appear at 718.2 eV, 720.6 eV, and 722.5 eV, indexing Fe^0^ 2p_1/2_, Fe^2+^ 2p_1/2_, and Fe^3+^ 2p_1/2_ [39]. Two typical satellite peaks of Fe at 715 and 724.2 eV were also observed [3]. These suggest the presence of metallic Fe and iron hydroxides/oxides. For the Co 2p spectrum (Figure 2b), the peaks appear at 777.4 eV, 780.4 eV, 793.2 eV, and 794.9 eV, which index to Co^0^ 2p_3/2_, Co^2+^ 2p_3/2_, Co^0^ 2p_1/2_, and Co^2+^ 2p_1/2_ separately [39]. Two typical satellite peaks are situated at 784.1 and 799.5 eV. Obviously, Co has only a zero and divalent valence state. In the high-resolution spectrum of Ni 2p (Figure 2c), two typical satellite peaks of Ni are found at 858.3 and 875.9 eV. Moreover, the peaks at 852.0 eV, 853.2 eV, 869.6 eV, and 871.9 eV conform to Ni^0^ 2p_3/2_, Ni^2+^ 2p_3/2_, Ni^0^ 2p_1/2_, and Ni^2+^ 2p_1/2_, separately, which demonstrates the presence of metallic Ni and divalent Ni compounds [18,21]. The high-resolution spectrum of Mo 3d (Figure 2d) displays the peaks located at 230.9 eV, 232.0 eV, 233.7 eV, and 235.2 eV, assigning to Mo^4+^ 3d_5/2_, Mo^6+^ 3d_5/2_, Mo^4+^ 3d_3/2_, and Mo^6+^ 3d_3/2_ separately [3]. The two peaks of the Zn 2p spectrum at 1020.8 and 1043.9 eV belong to Zn^2+^ 2p_3/2_ and Zn^2+^ 2p_1/2_ separately (Figure 2e) [40]. In the O 1s spectrum, the peaks at 531.8 and 533.1 eV indicate M-OH and H_2_O (Figure 2f) [41]. The intense signal of M-OH implies that the synthesized sample is mainly composed of hydroxides, which is also confirmed by the conclusion of XRD analysis (Figure 1c).

To understand the coordination environment of FeCoNiMoZn-OH HEH, Raman spectroscopy measurement was performed (Appendix A). The prominent Raman signals at about 460 and 530 cm^−1^ are derived from the existence of metal-hydroxide (M-OH) bonds [42]. In addition, the spectral features of FeCoNiMoZn-OH HEH are relatively broad and diffuse. This demonstrates the lower crystallinity of the obtained catalysts and the higher density of lattice defects [43]. FTIR spectrum revealed the presence of oxygen-containing groups in FeCoNiMoZn-OH HEH (Appendix A). The peaks at 3380  cm^−1^ and 1650 cm^−1^ indicate the hydroxyl radicals and the interlayer water, separately [44]. These are in accordance with the findings of XRD and XPS.

The OER catalytic activity of the obtained samples was revealed by LSV curves tested in O_2_-saturated 1.0 M KOH aqueous solution. It is found that FeCoNiMoZn-OH HEH demonstrates the most excellent OER catalytic activity compared to FeCoNiMo-OH, FeCoNiZn-OH, FeCoNi-OH, RuO_2_, and NF samples (Figure 3a). Figure 3b shows the histogram of overpotential at different current densities. FeCoNiMoZn-OH HEH possesses the lowest overpotential of 248 mV at the current density of 10 mA cm^−2^ in comparison to FeCoNiMo-OH (267 mV), FeCoNiZn-OH (287 mV), and FeCoNi-OH (280 mV) catalysts and commercial RuO_2_ (261 mV). Moreover, FeCoNiMoZn-OH HEH also has the lowest overpotential of only 269 mV at 50 mA cm^−2^ among the obtained samples, followed by FeCoNiMo-OH (310 mV), FeCoNi-OH (328 mV), and FeCoNiZn-OH (332 mV) catalysts and commercial RuO_2_ (340 mV). Obviously, the introduction of Zn almost has no effect on the overpotential of FeCoNiZn-OH catalyst compared to FeCoNi-OH. In contrast, the introduction of Mo significantly lowers the overpotential of FeCoNiMo-OH, suggesting that Mo can promote the catalytic efficiency of OER. Although Zn has little effect on lowering the overpotential of FeCoNiZn-OH, its introduction further reduces the overpotential of FeCoNiMoZn-OH. This demonstrates the positive synergistic effect of Mo and Zn in FeCoNiMoZn-OH HEH. The Tafel slope is a significant indicator to reflect the kinetic characteristics of the OER-catalyzed process. Figure 3c shows that FeCoNiMoZn-OH HEH has the smallest tafel slope of 30 mV dec^−1^ in comparison to FeCoNi-OH (53 mV dec^−1^), FeCoNiMo-OH (42 mV dec^−1^), FeCoNiZn-OH (45 mV dec^−1^), and RuO_2_ (108 mV dec^−1^), demonstrating the superior OER kinetics of FeCoNiMoZn-OH HEH.

Figure 3d displays the typical Nyquist plots of the obtained catalysts and the corresponding equivalent electrical circuit. R_ct_ and R_s_ represent charge transfer resistance and electrolyte solution resistance, separately. Among all obtained electrocatalysts, FeCoNiMoZn-OH HEH displays the smallest diameter of the semi-arc of R_ct_, demonstrating its superior properties in charge transfer. This is conducive to the acceleration of OER and therefore improves the catalytic activity. Additionally, the CV curves of the catalysts tested at various scan rates were collected (Appendix A) to estimate the ECSA of the obtained catalysts. ECSA is determined by C_dl_ (ECSA = C_dl_/C_s_, the C_s_ was set as 40 μF cm^−2^) [21]. As shown in Figure 3e, FeCoNiMoZn-OH HEH electrocatalyst displays the largest C_dl_ value (2.35 mF cm^−2^) among the obtained catalysts, almost two times higher than the FeCoNi-OH catalyst (1.30 mF cm^−2^). Correspondingly, the FeCoNiMoZn-OH HEH has the largest ECSA of 58.8 cm^2^ compared with FeCoNiMo-OH (37.5 cm^2^), FeCoNiZn-OH (37.0 cm^2^), and FeCoNi-OH (32.5 cm^2^) catalysts (Appendix A), suggesting the most potential active sites of FeCoNiMoZn-OH HEH. This is ascribed to the abundant structural defects of FeCoNiMoZn-OH HEH.

The LSV curves normalized by ECSA are employed to investigate the intrinsic activity of the obtained electrocatalysts. FeCoNiMoZn-OH HEH has the highest intrinsic activity among the obtained samples, followed by FeCoNiMo-OH, FeCoNi-OH, and FeCoNiZn-OH catalysts (Appendix A). The introduction of Zn results in a slight reduction in the intrinsic activity of FeCoNiZn-OH eletrocatalyst compared to FeCoNi-OH. This may be attributed to the fact that Zn itself is catalytically inactive and occupies some active sites or covers the active sites. On the contrary, the introduction of Mo plays a vital role in improving the intrinsic activity of the FeCoNiMo-OH HEH catalyst. As a high valence metal, Mo allows fast multi-electron transfer between the species with multiple oxidation valence states, favoring the improvements in catalytic intrinsic activity [45]. Interestingly, the combination of inactive Zn and Mo in FeCoNiMoZn-OH HEH produces a positive synergistic effect between Zn and Mo, which results in the further improvement in the intrinsic activity of FeCoNiMoZn-OH HEH. It is possible that the Zn can effectively tune the amounts of Mo^4+^ and Mo^6+^ and improve the distribution of high valence ions, thus regulating the adsorption process of the OER intermediates.

In addition, the mass of the prepared samples before and after electrodeposition were measured and the loading of the obtained catalysts was compared (Table 1). There was little difference in the loading (∆M) of the prepared catalysts. Moreover, the turnover frequency (TOF) of FeCoNiMoZn-OH HEH is calculated assuming that all materials obtained by deposition are active sites. Notably, the TOF of the FeCoNiMoZn-OH HEH is as high as 1.81 s^−1^ at 10 mA cm^−2^ (Appendix A), suggesting an enhanced electrocatalytic oxygen production capability. Compared with other reported advanced catalysts, the obtained FeCoNiMoZn-OH HEH exhibits a high TOF (Appendix A).

Moreover, the long-term durability of catalysts is regarded as another crucial indicator of catalyst performance. Figure 3f shows the chronopotentiometry curve of the FeCoNiMoZn-OH HEH tested in an alkaline aqueous solution. The OER polarization potential was consistently maintained at about 1.51 V (vs. RHE) for 55 h at 10 mA cm^−2^, demonstrating its satisfactory long-term durability. To reveal the change in morphology, structure, and composition of FeCoNiMoZn-OH HEH after stability testing for 55 h, a series of characterizations of the obtained samples were carried out. After stability testing for 55 h, FeCoNiMoZn-OH HEH remains the integrated 3D porous morphology consisting of plenty of nanosheets (Figure 4a–c), demonstrating its excellent structural stability. The elements of Fe, Co, Ni, Mo, Zn, and O are still observed and uniformly distributed in FeCoNiMoZn-OH HEH (Figure 4d), further implying excellent OER durability of FeCoNiMoZn-OH HEH.

## 4. Comparison with the Literature

Table 2 shows the comparison of FeCoNiMoZn-OH HEH with the state-of-the-art catalysts reported in previous work. The obtained FeCoNiMoZn-OH HEH exhibits the extraordinary catalytic activity of OER (overpotential of 240 mV at 10 mA cm^−2^, Tafel slope of 30 mV dec^−1^) in 1 M KOH aqueous solution compared to the catalysts reported in previous work.

## 5. Conclusions

In summary, a novel FeCoNiMoZn-OH-layered HEH was synthesized by electrodeposition under a mild atmosphere. The obtained FeCoNiMoZn-OH exhibits 3D porous nanosheet morphology and a polycrystalline structure. This structure possesses abundant lattice defects and a large number of surface unsaturated coordination active atoms, providing plenty of active sites. The FeCoNiMiZn-OH HEH demonstrates excellent OER catalytic performance under alkaline conditions, including low overpotential (248 mV at 10 mA cm^−2^) and outstanding durability. The positive synergistic effect of Mo and Zn optimizes the electronic structure of HEH and thus modulates the adsorption process of intermediates, enhancing the intrinsic activity for OER. The obtained catalyst in this work exhibits excellent catalytic performance under laboratory conditions and is expected to be practically applied to the electrolysis of water. This study will afford an efficient strategy for the design and synthesis of novel high-entropy catalysts with high performance.

## Figures and Tables

**Figure 1 nanomaterials-14-00889-f001:**
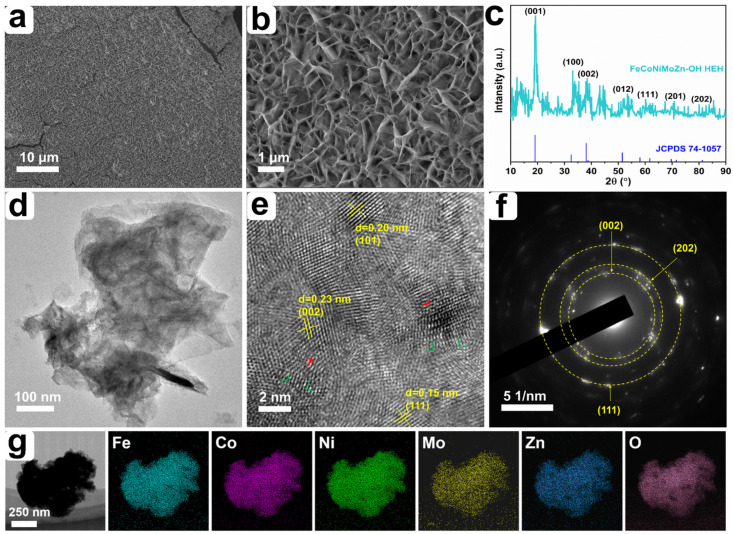
(**a**,**b**) SEM images, (**c**) XRD spectrum, (**d**) TEM image, (**e**) HRTEM image, (**f**) the corresponding SAED pattern, and (**g**) elemental mappings of FeCoNiMoZn-OH HEH.

**Figure 2 nanomaterials-14-00889-f002:**
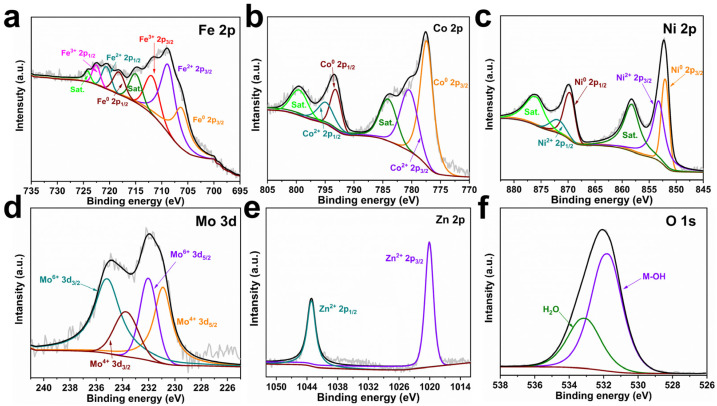
XPS high-resolution spectra of FeCoNiMoZn-OH HEH. (**a**) Fe 2p, (**b**) Co 2p, (**c**) Ni 2p, (**d**) Mo 3d, (**e**) Zn 2p, and (**f**) O 1s spectra of FeCoNiMoZn-OH HEH.

**Figure 3 nanomaterials-14-00889-f003:**
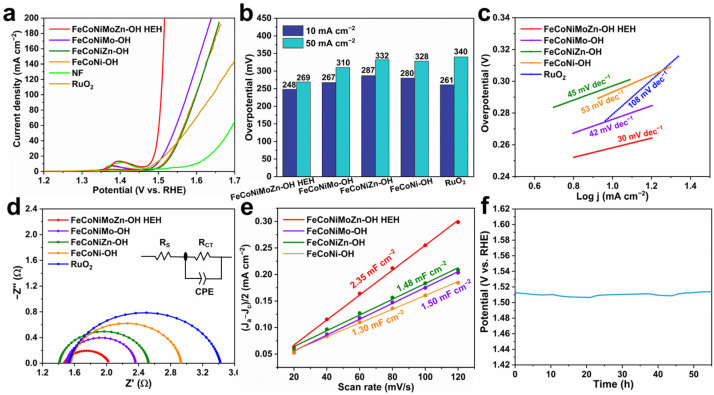
Electrocatalytic performance of FeCoNiMoZn-OH electrocatalysts. (**a**) LSV, (**b**) histogram of overpotential, (**c**) Tafel slope, (**d**) EIS, (**e**) C_dl_ values, and (**f**) chronopotentiometry curve.

**Figure 4 nanomaterials-14-00889-f004:**
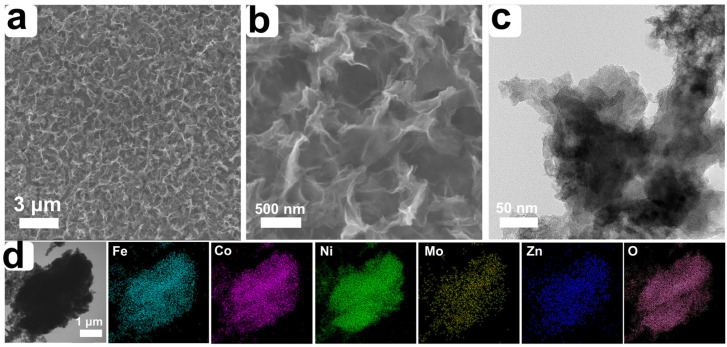
(**a**,**b**) FESEM images, (**c**) TEM image, and (**d**) corresponding elemental mappings of FeCoNiMoZn-OH HEH after stability testing for 55 h.

**Table 1 nanomaterials-14-00889-t001:** Comparison of the mass of the prepared samples before and after electrodeposition.

Materials	M_1_ (mg)	M_2_ (mg)	∆M (mg)
FeCoNiMoZn-OH HEH	52.5	55.7	3.2
FeCoNiMo-OH	53.7	56.6	2.9
FeCoNiZn-OH	51.6	55.1	3.5
FeCoNi-OH	52.6	56.4	3.8

M_1_ is the mass of bare NF and M_2_ is the mass of the sample after electrodeposition. ∆M indicates the difference in mass before and after deposition.

**Table 2 nanomaterials-14-00889-t002:** Comparison of OER performance of FeCoNiMoZn HEH catalyst with the state-of-the-art electrocatalysts.

Catalysts	Electrolyte	*η*_10_ (mV)	Tafel Slope (mV dec^−1^)	Synthesis Method	Ref.
Co_3_O_4_-MoSe_2_@C	1 M KOH	360	75.7	hydrothermal method	[46]
Co-Mo_2_C@NC	1 M KOH	440	156	solvothermal method	[47]
Co_2_B/CoSe_2_	1 M KOH	320	56	wet chemical method	[48]
Ce-MnCo_2_O_4_	1 M KOH	390	125	coprecipitation and calcination	[49]
La_0.5_Sr_0.5_Mn_0.15_Fe_0.15_Co_0.4_Ni_0.15_Cu_0.15_O_3_	1 M KOH	309	93.4	electrospinning technology	[50]
MnFeCoNi HEA	1 M KOH	302	83.7	Mechanical alloying	[51]
(Fe_0.2_Co_0.2_Ni_0.2_Cr_0.2_Mn_0.2_)_3_O_4_	1 M KOH	275	50.3	solution combustion	[52]
CoFeNiMnMoPi	1 M KOH	270	74	high-temperature fly-through method	[53]
(CoNiMnZnFe)_3_O_3.2_	1 M KOH	336	47.5	mechanical alloying	[54]
FeCoNiCuIr NPs	1 M KOH	360	70.1	heat-up method	[55]
(Cr_0.2_Mn_0.2_Fe_0.2_Ni_0.2_Zn_0.2_)_3_O_4_	1 M KOH	295	53.7	solvothermal method	[56]
Cu_0.5_Fe_0.5_NNi_2_Co_0.5_Fe_0.5_	1 M KOH	370	55	chemical solution deposition	[57]
AlNiCuCoFeY	1 M KOH	261	50	melt spinning	[58]
FeCoNiMnCu	1 M KOH	280	59	electrodeposition	[39]
FeCoNiMnW	1 M KOH	512	161	electrodeposition	[4]
(Fe_0.2_Co_0.2_Ni_0.2_Mn_0.2_Cr_0.2_)_3_O_4_@CC	1 M KOH	287	95.8	electrodeposition	[59]
CNFMPO	1 M KOH	252	44.3	electrodeposition	[60]
This work	1 M KOH	248	30	electrodeposition	−

## Data Availability

Data are contained within the article.

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
