# Peer review of "Novel High-Entropy FeCoNiMoZn-Layered Hydroxide as an Efficient Electrocatalyst for the Oxygen Evolution Reaction"

_nanomaterials, 2024, doi:10.3390/nano14100889_

Round 1
Reviewer 1 Report
Comments and Suggestions for Authors
In this work, a complex multi-component catalyst for hydrogen evolution reaction was prepared.
1. There is no detailed description of the synthesis methodology. Were the solutions doped or immediately drained?
2. There is no method to confirm the composition of the obtained catalyst
3. How many experiments were performed? What is the variation of results?
4. The catalysts obtained are not sufficiently characterized. Raman spectroscopy should be used to determine the number of defects and estimate interplanar distances. IR spectroscopy is also needed to prove the composition of the catalyst.
5. Approaches to ensure consistency of catalyst composition are not given.
6. XPS contains only metal peaks and metal-water bonds, so it does not confirm the composition.
7. No monocomponent catalysts are given so that the synergistic effect of the resulting catalyst can be evaluated
Comments on the Quality of English Language
Extensive editing of English language required
Reviewer 2 Report
Comments and Suggestions for Authors
Title: Novel High-Entropy FeCoNiMoZn Layered Hydroxide as an Efficient Electrocatalyst for Oxygen Evolution Reaction
This paper aims to evaluate the performance of FeCoNiMoZn layered hydroxide for OER and requires careful revision for publication.
Comments:
- Clarify the novelty of your materials. What sets them apart from existing ones?
- Why did the authors opt not to use the first five 3D elements (Sc, V, Ti, Cr, and Mn) in their selection process?
- What was the rationale behind not selecting copper for this study?
- Incorporate details in the introduction about the rationale for selecting specific elements to create high-entropy materials for OER.
- Consider potential complications or errors introduced by using nickel foam as a substrate. Experimental evidence to support its suitability is needed.
- Provide details about the nickel foam's size, weight, thickness, and pore size.
- Include the weight of deposited materials on nickel foam in a table for clarity.
- Address whether the materials have been uniformly deposited on both sides of the nickel foam.
- Provide the EDS layered image separately and the EDS line spectrum.
- Line 157: "This unique 3D porous structure not only provides a large active surface area but also contributes to ions/mass transportation." How can the authors report the active surface area using SEM?
- Line 170: "Is your material polycrystalline?
Reviewer 3 Report
Comments and Suggestions for Authors
In this work, Z. Cheng et al. successfully constructed high-entropy FeCoNiMoZn layered hydroxide on nickel foam via a electrochemical deposition method under mild conditions, obtaining a catalyst exhibiting very high OER performance, as demonstrated by the overpotential value of 248 mV obtained at 10 mA cm-2 and very low Tafel slope of 30 mV dec-1. The manuscript introduces novelty in the field and could be appealing to the Nanomaterials journal readership. Hence, it could be considered for publication in Nanomaterials after addressing the following major revisions:
1. It would be interesting if authors could add also the 20 CV cycles for activation of catalysts (perhaps in ESI). What is the rational to stop the activation after the 20th cycle?
2. SEM reported in Fig. 1a displays some surface cracks, could they be detrimental for the OER activity?
3. In the XRD reported in Fig.1c, it is suggested to add in the figure also the (012), (111), (201), and (202) reflections of FeCoNiMoZn-OH phase.
4. No evidence of turnover frequency (TOF) values is provided in the text among all catalysts. However, TOF value is important to elucidate the intrinsic activity of a catalyst, since it is independent from mass loading. Authors are suggested to provide TOF values, if present.
5. To further emphasize the value and novelty of this work, authors are suggested to compare the OER activity reported in table S1 with other similar HEMs catalysts, indicating not only the overpotential and Tafel slope but also the fabrication method and temperature used in these studies, since in the introduction they claim that other methods of preparation of HEMs involve hash high-temperature conditions.
6. Are the LSV reported in Fig. 3a corrected for iR drop? It is not specified in the experimental section.
7. In Fig.3b histogram reports the overpotential at 20 mA cm-2; however this is not correspondant with the text, indeed the authors in the text report the overpotential at 50 mA cm-2, please correct.
8. Authors used Ag/AgCl as reference electrode, Hg/HgO is more stable in KOH media, then it should be preferrable.
9. On pag.7, line 261, authors affirm that hey performed structural characterization after stability testing for 55h. However, it is reported only the morphological and compositional characterization by SEM and TEM. Authors are suggested to add also XRD characterization post stability test, for sake of completeness.
10. Do the authors envision a possible application of this high entropy catalyst also for HER, perhaps in other material phase?
11. In the conclusions section author report only the experimental outcomes of this work. I would suggest to add also a general sentence in the end where they highlight the impact of this work on the scientific community and some perspectives.
12. The authors are advised to cite in the introduction recent published relevant articles focused on transition metal based electrocatalysts for water electrolysis, such as Catalysis Today 2023, 423, 113929 (https://doi.org/10.1016/j.cattod.2022.10.011), ACS Catal. 2022, 12, 17, 10808–10817 (https://doi.org/10.1021/acscatal.2c02604), and others.
Reviewer 4 Report
Comments and Suggestions for Authors
In the submitted manuscript, FeCoNiMoZn mixed phase was produced on nickel foam by the electrochemical deposition method. The obtained material was tested as an electrocatalyst in the oxygen evolution reaction. The work could be accepted for publication in Nanomaterials, but requires major revision in accordance with the suggested points below.
1. In Fig. 1 SEM images of unmodified nickel foam should also be shown for comparison.
2. The work absolutely lacks a quantitative analysis of the deposited metals - Fe, Co, Ni, Mo and Zn. Of course, I am aware of the difficulties in determining the correct nickel content, but the remaining components should not be a problem.
3. Looking at the diffraction pattern shown in Fig. 1c, it is difficult to clearly conclude the existence of the postulated FeCoNiMoZn-OH HEH phase. Even if it occurs, it is in a very fine-crystalline form and certainly not the only crystalline phase. The discussion needs to be changed in this regard. By the way, how was the diffraction pattern collected? If the created film was not separated from the nickel foam surface, shouldn't reflections from the latter component be observed?
4. The peak fitting in the XPS Co 2p spectrum on the side of lower binding energies does not correspond to the baseline at all.
5. Moreover, the baseline in the XPS Zn 2p spectrum has a surprising shape. Let me just remind that the baseline characteristic measured in XPS should be non-linear and decrease in value as the binding energy decreases.
6. The sources used to interpret the XPS results are not presented.
7. The origin of nickel foam is not given in Experimental.
8. Check the nickel sulfate formula, as it seems to be incorrectly specified.
Comments on the Quality of English Language
The manuscript must be revised because contains a lot of spelling and grammatical errors.
Round 2
Reviewer 1 Report
Comments and Suggestions for Authors
The study could be accepted in present form. However, moderate editing of English language required before publication
Comments on the Quality of English Language
Moderate editing of English language required
Author Response
Thank you for your valuable comments. We have revised the English language of the manuscript appropriately to meet the requirements.
Reviewer 2 Report
Comments and Suggestions for Authors
Dear authors,
Thank you so much for considering the reviewer's comments on improving the manuscript. However, the reviewer requested the improvement as below.
1. Please go through the ithenticate results (37%). Authors should minimize it to 15 to 20%. Otherwise, it should not be recommended for publication.
2. Include the details about reaction conditions instead of the mild atmosphere in line 15.
3. Comments 2: Please include your clarification in the introduction section.
4. Comments 7: Please include the table in the main text.
5. Please include the number of repeatable experiments.
6. Comments 9: Please include the EDS line spectrum in Figure 2 and Figure 4 with the atomic weight percentage of elements.
7. Please add a "comparison with literature" section before the conclusion section and include Table S1 here. Table S1 needs modification with a separate reference column and should include or replace findings with samples synthesized by electrodeposition. Then, explain the findings after the Table.
8. Please include the limitations of this study in the conclusions.
Reviewer 3 Report
Comments and Suggestions for Authors
The authors responded well to the comments, I can recommend the publication of the paper in the present form.
Author Response
Thank you for reviewing the manuscript and for your valuable suggestions!
Reviewer 4 Report
Comments and Suggestions for Authors
All my comments have been considered and the manuscript has been corrected accordingly.
Author Response
Thank you for your efforts in reviewing this manuscript and for your comments!